

**Forecasting post-failure landslide mobility using a SPH model**
**and data from ring shear strength tests: A case study**
Miao Yu[1], Yu Huang[2,3]*, Wenbin Deng[2], Hualin Cheng[2]
[1] Faculty of Engineering, China University of Geosciences, Wuhan, Hubei 430074,
China
[2] Department of Geotechnical Engineering, College of Civil Engineering, Tongji
University, Shanghai 200092, China
[3] Key Laboratory of Geotechnical and Underground Engineering of the Ministry of
Education, Tongji University, Shanghai 200092, China
* Corresponding author: Yu Huang (Tel.: +86-21-6598-2384; Fax: +86-21-6598-5210.
E-mail: yhuang@tongji.edu.cn)
**Abstract** Flowlike landslides, such as flowslides and debris avalanches, have caused
serious infrastructure damage and casualties for centuries. Effective numerical
simulation incorporating accurate soil mechanical parameters is essential for
predicting post-failure landslide mobility. In this study, smoothed particle
hydrodynamics (SPH) incorporating soil ring shear test results was used to forecast
the long-runout mobility for a landslide on an unstable slope in China. First, a series
of ring shear tests under different axial stresses and shear velocities were conducted to
evaluate the residual shear strength of slip zones after extensive shear deformation.
Based on the ring shear test results, SPH modeling was conducted to predict the
post-failure mobility of a previously identified unstable slope. The results indicate that





the landslide would cut a fire road on the slope after 12 s and cover an expressway at
the foot of that slope after 36 s. In the model, the landslide would finally stop sliding
about 25 m beyond the foot of the slope after 120 s. This study shows that combining
the SPH method with ring shear test results to forecast landslide mobility can provide
basic information for landside disaster mitigation.
**Keywords:** Landslide hazard; Post-failure mobility; Ring shear tests; Smoothed
particle hydrodynamics (SPH); Residual strength
**1. Introduction**
Flowlike landslides triggered by intense earthquakes or rainfall, such as debris and
rock avalanches, have caused serious infrastructure damage and casualties for
centuries (Okada et al. 2007; Wang et al. 2005). This kind of landslide is commonly
high-speed and has a long runout distance. For example, a large landslide in southern
Italy in February, 2010, had a runout distance of 1.2 km and necessitated the
evacuation of nearly 2,300 people. This landslide was triggered by heavy and
prolonged rainfall between August 2009 and February 2010 (Gattinoni et al. 2012).
The 2009 Shiaolin landslide in Taiwan, induced by a cumulative rainfall of nearly
1700 mm from Typhoon Morakot, buried Shiaolin Village and resulted in more than
400 people dead and missing (Tsou et al. 2011). Numerical simulations that
incorporate accurate soil mechanical parameters are a powerful tool for simulating
landslide runout distances; these simulations can provide fundamental reference
information for landside disaster mitigation (Yerro et al. 2016; Žic et al. 2015).



The main numerical methods for simulating landslides are the discrete element
methods and the continuum methods (Lu et al. 2014; Wu et al. 2017). Using a discrete
element method, such as the distinct element method (DEM) or discontinuous
deformation analysis (DDA), the nonphysical parameters cannot be determined
exactly (Huang et al. 2014). However, continuum methods based on grids, like the
finite element method (FEM) and the finite difference method (FDM), have the
shortcomings of grid distortion and low accuracy for the numerical analysis of a
landslide with a long runout. Recently, a new numerical method has been used to
overcome these limitations, namely the smoothed particle hydrodynamics method
(SPH) (Bui et al. 2008). This method is in the framework of continuum methods. SPH
is a pure Lagrangian, meshless hydrodynamics method and it is capable of simulating
flow deformation, free surfaces, and deformation boundaries (Liu and Liu 2003).
Several studies have demonstrated the efficiency of the SPH method for the large
deformation analysis post landslide. Huang et al (2014) provided a general view of
SPH applications for solving large deformation and failure problems such as dam
breaks, slope failure, and soil liquefaction flow. Pastor et al (2009) applied a
depth-integrated, coupled SPH model successfully to simulate catastrophic flow-like
landslides that occurred in southern Italy in 1998. Cascini et al (2014) proposed a
SPH model to represent two actual flow-type events accurately. Cuomo et al (2016)
used SPH to simulate flow-like landslides (debris flows and debris avalanches) and
discussed the influence of bed entrainment on landslide propagation. Hu et al (2015)



conducted two- and three-dimensional SPH numerical simulation of flow-like
landslides triggered by the 2008 Wenchuan earthquake in China and proposed that the
SPH method is well-suited for modeling free surfaces, moving interfaces, and
extensive deformation.

Study into the residual shear strength property of slip zones under large shear

deformation is essential to landslide long-runout mechanism explanation (Tika and
Hutchinson 1999; Wen et al. 2007). Because the physical sample displacement using
conventional laboratory shear tests, like direct shear tests and triaxial shear tests, is
limited to about 10 mm (Okada et al. 2007; Casagli et al. 2006; Van Asch et al 2007),
the shear behavior for large shear displacements cannot be assessed by these methods
(Dai et al. 2016). Ring shear tests, which can impart extremely large shear strains, may
be the ideal laboratory tool for extensive shear deformation testing (Okada et al. 2007;
ASTM Standard D7608-10, 2010). Several studies have applied ring shear tests to
study the residual shear strength of soils (Fukuoka et al, 2007; Hoyos et al. 2014; Li et
al. 2013; Wang et al. 2005). For example, Fukuoka et al (2007) applied a newly
developed ring shear test to study shear zone development during large displacements.
That study pointed out that a ring shear test is the most appropriate test for studying
long-travel landslides. Kimura et al (2014) studied the effect of the shearing rate on
the residual strength of landslide soils using ring shear tests. Zhang et al (2011) used
ring shear tests to study the transform mechanism of the slide-debris flow under large



deformation. Li et al (2017) explored the residual strength of silty sand under different
degrees of over consolidations and different shear rates using ring shear tests.

This study presents an effective numerical simulation method, namely SPH, that

incorporates accurate soil mechanical parameters derived from ring shear tests. The
aim is to predict the downslope flow after slope failure of a previously identified
unstable slope and thereby provide basic information for landside disaster mitigation.
First, this paper describes the geomorphological and geological setting, hydrogeology
and rainfall, and triggering factors of the landslide examined for this case study.
These descriptions are based on detailed fieldwork. Next, a series of ring shear tests
under several different normal stresses and shear rates were performed to identify the
shear strength of the landslide soil. Finally, a SPH-based numerical simulation of the
landslide was run to predict the extent of the landslide and track the slide velocity at
different times.
**2. A case study – the Dafushan landslide**
*2.1 Geomorphological and geological setting*
The Dafushan landslide, located in the Panyu District, Guangzhou City, South China,
was selected for this case study (Fig. 1(a)). The slope is primarily composed of
Cretaceous silty mudstone, conglomerate, and sandstone overlain by Quaternary silty
clay (Yu et al. 2017) (Fig. 1(b)). The landslide is creeping from the northeast to the
southwest covering an area of about 70 m × 40 m (Fig. 1(c)). The height difference
between the toe and the crown is approximately 20 m with an average gradient of 25°.

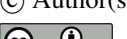
The Dongxin expressway and a 50 t, high-voltage power line tower are located at the
toe and top of the slope, respectively. In addition, there is a fire response service road
that runs along the slope that is affected by the slide (Fig. 1(d)).

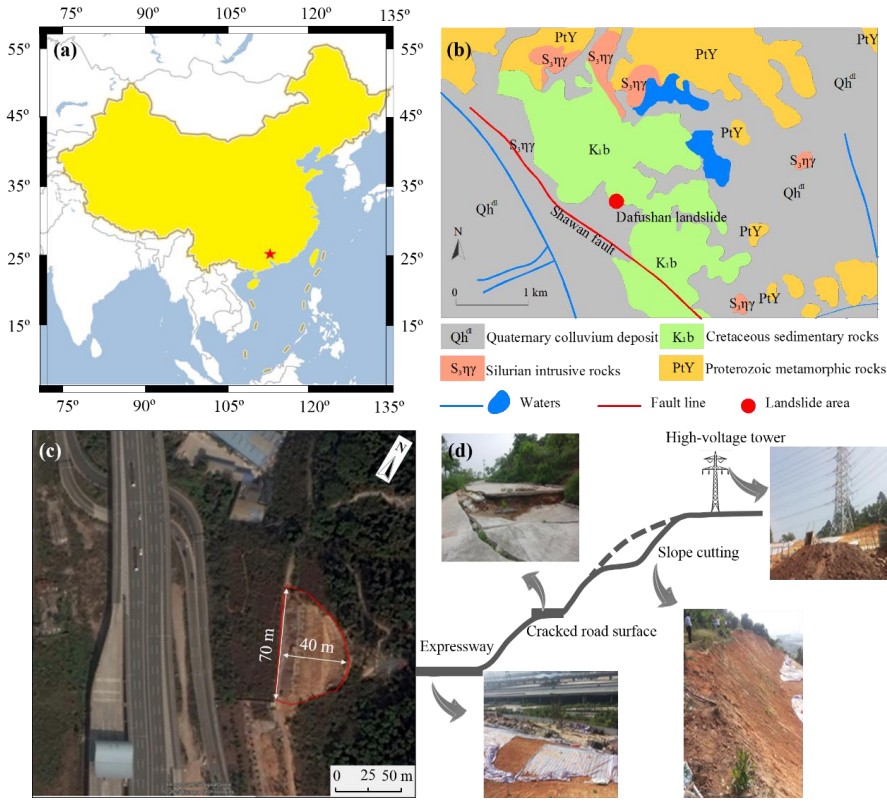

**Fig. 1.** Overview of the Dafushan landslide. (a) Landslide location; (b)
Geomorphologic and geologic map of the landslide area; (c) Aerial view of the
unstable (potential landslide area (the area inside the red lines) (image from Google
Earth®); (d) Engineering activities on the slope (reprinted from Yu et al. (2017) with
permission of Springer).
*2.2 Landslide triggering factors*





The ground was first found to be unstable in May 2013. This instability was
manifested mainly by cracks in the ground surface and cracks in the
round-the-mountain road. The road was built for fire response services in May 2011.
The relevant departments repaired the damaged road immediately to guarantee the
normal operation of the road. However, addition evidence of instability was found in
the middle of August 2013 after a period of intense rainfall. The road was damaged
again and the trees up the hill began to tilt. Based on preliminary field investigation,
the main factors that triggered the landslide were deduced.
*(1) Hydrogeology and rainfall*
Rainfall is the main supply source of groundwater in the study area. The average
annual rainfall is 1635.6 mm. Most of the rain falls between April and September; this
rainfall accounts for 81% of the yearly precipitation. In the rainy season, the
groundwater level rise significantly and reduces the shear strength of the soil.
Combined with the rainfall flushing effect on the slope surface, the stability of the
slope is decreased significantly.
*(2) Mechanical properties of landslide soil*
The shallow part of the landslide is mainly composed of silty clay (Fig. 2) and a
strongly weathered mudstone soil with a low shear strength. These materials soften
and disintegrate when wet, thus the slope is stable in the dry season but shows signs
of instability in the rainy season.



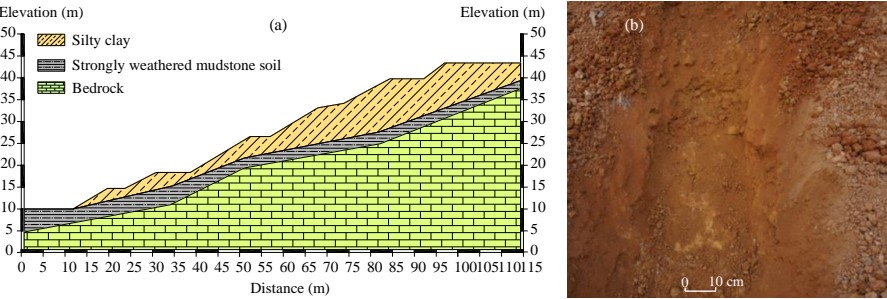

**Fig. 2.** Geology and soil at the Dafushan landslide. (a) Longitudinal geologic section

of the unstable slope shown in Fig. 1(c). (b) Photograph of the silty clay landslide soil.

*(3) Human engineering activities*

Human engineering activities impaired the natural stability of the slope. Two

examples: a) to build the fire service road, a cut was made in the slope; b) the heavy

high-voltage power line tower increases the downward pressure on the slope (Fig.

1(d)).

**3. Ring shear tests**

A GCTS Residual Ring Shear Testing System (model SRS-150) produced by

Geotechnical Consulting and Testing Systems (GCTS) in 2012 in the USA was used

for the ring shear tests conducted for this study (Fig. 3). The SRS 150 is a fully

automated electro-pneumatic and servo-controlled testing system used for

determining the residual strength of continuously sheared soil. Shear torques of up to

820 Nm can be applied, consolidation stress can be up to 1000 kPa, and unlimited

angular rotation is allowed (Dai et al. 2016; Hoyos et al. 2014). The unit is capable of

applying shearing rates of 0.001 to 360 degrees per minute continuously with





zero-backlash for replication of true in-situ strain rates during failure. (Hoyos et al.

2011).

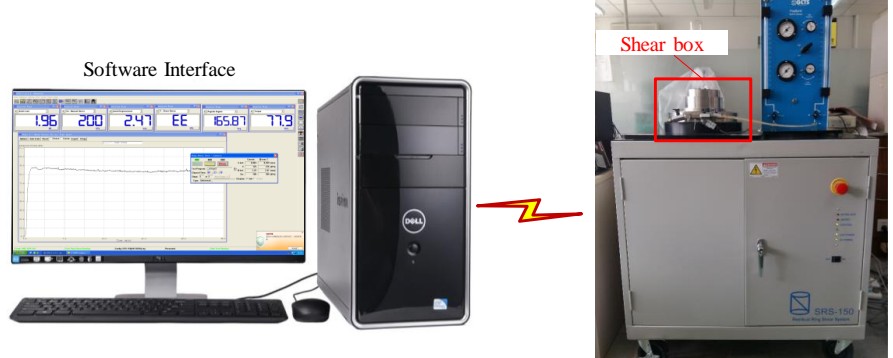


**Fig. 3.** Photograph of the GCTS SR-150 Residual Ring Shear testing device and an
image of the GCTS software interface.
A schematic illustration of a sample in the apparatus is shown in Fig. 4. For
testing granular materials, the device accepts ring-shaped samples with a 150 mm
outer diameter and a 100 mm inner diameter. The sample is sheared by rotating the
upper half of the testing unit and keeping the lower half motionless. Two types of
shearing modes, either a shear speed control mode or a shear torque control mode, can
be chosen.




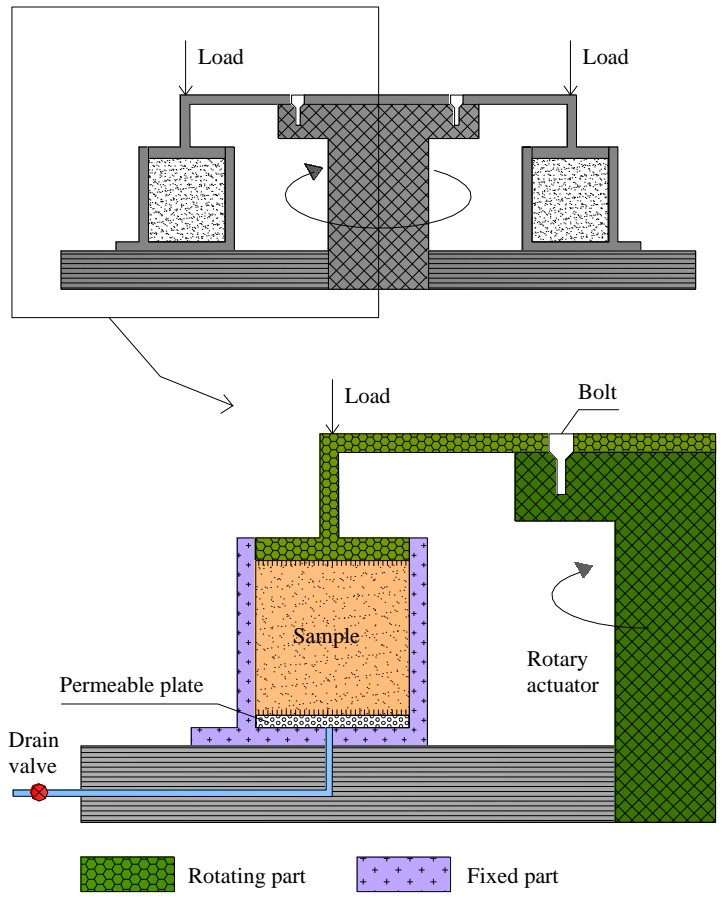


**Fig. 4.** Schematic cross sections of ring shear apparatus shown in Figure 3.

*3.1 Sample preparation and test procedures*
The samples studied were samples of the silty clay soil from the Dafushan landslide
shown in Fig. 2(b). The soil's physical properties are listed in Table 1.

**Table 1** Physical properties of a soil from the Dafushan landslide.

| Density $\rho$ (g/cm³) | Dry density $\rho_d$ (g/cm³) | Water content $\omega$(%) | Liquid limit $\omega_L$ | Plastic limit $\omega_P$ | Plastic index $I_P$ | Liquidity index $I_L$ |
|---|---|---|---|---|---|---|



| | | | (%) | (%) | | |
|---|---|---|---|---|---|---|
| 1.77 | 1.43 | 21.4 | 29.8 | 17.5 | 12.3 | 0.32 |

A series of ring shear tests were performed to determine the physical properties
of the landslide soil after it had been extensively sheared. The saturated soil sample
was first consolidated under a normal stress and then it was sheared to a residual state
under naturally drained conditions using the shear speed control mode of the ring
shear test system. For these tests, normal stresses of 50, 100, 200, 300, and 400 kPa
were used to consolidate the soil samples and different shear rates (1, 5, 10, 20 °/min)
were employed. Test parameters are listed in Table 2.
**Table 2** Consolidation stresses, shearing rates, and saturations for soil specimens subjected to
laboratory ring shear tests.

| Test | Normal stress σ (kPa) | Shear rate α (°/min) | Saturation (%) |
|---|---|---|---|
| 1-1 | 50 | 5 | 100 |
| 1-2 | 100 | 5 | 100 |
| 1-3 | 200 | 5 | 100 |
| 1-4 | 300 | 5 | 100 |
| 1-5 | 400 | 5 | 100 |
| 2-1 | 200 | 1 | 100 |
| 2-2 | 200 | 5 | 100 |
| 2-3 | 200 | 10 | 100 |



| 2-4 | 200 | 20 | 100 |
|-----|-----|----|-----|
| 3-1 | 50 | 5 | 0 |
| 4-2 | 100 | 5 | 0 |
| 3-3 | 200 | 5 | 0 |
| 3-4 | 300 | 5 | 0 |
| 3-5 | 400 | 5 | 0 |

*3.2 Test results and discussion*
(1) Axial stress
Figure 5 shows the relationships between shear stress and angular displacement
under a shear rate of 5 °/min and axial stresses of 50, 100, 200, 300, and 400 kPa. At
the same shear rate, shear strength increases with increasing axial stress. In the initial
shear stages, shear stresses increase rapidly along with shear displacement and reach a
peak shear strength. The greater the axial stresses, the larger the shear displacement at
peak shear strength. When the axial stress is low (e.g., 50 kPa and 100 kPa), the shear
stresses do not change after peak shear strength is reached. When the axial stress is
high (e.g., 200 kPa, 300 kPa, or 400 kPa), the shear stresses decrease after peak shear
strength but eventually stabilize. This stable strength is the residual shear strength and
is the result of strain softening.


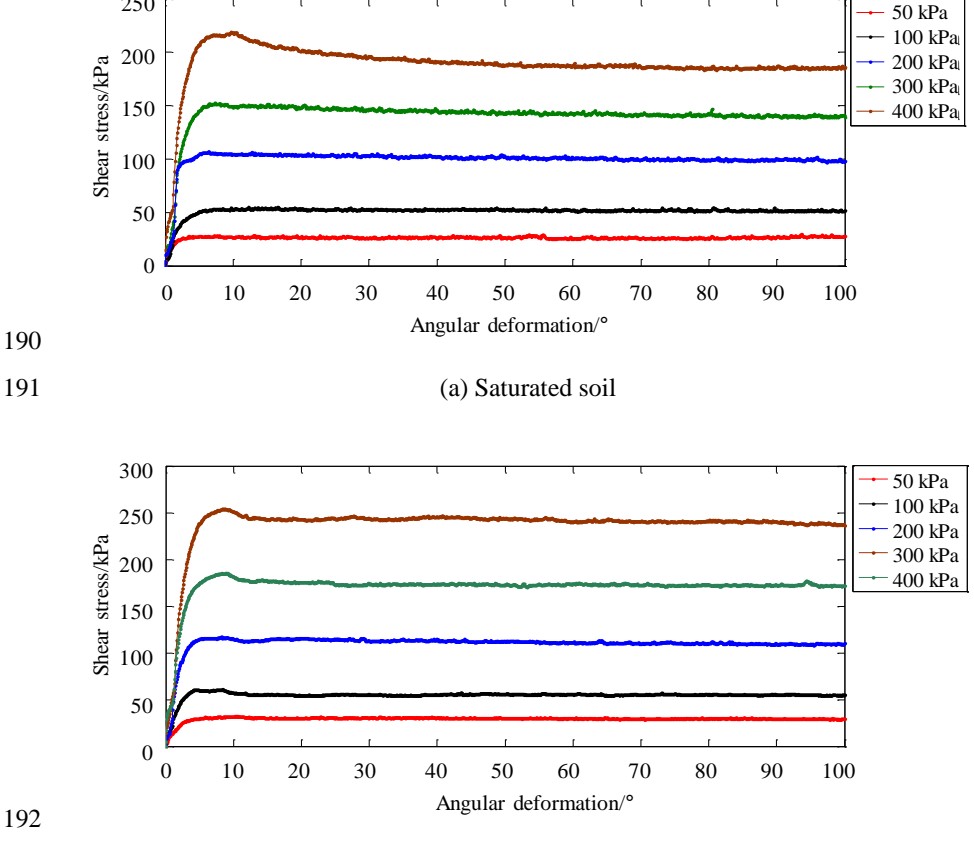


(a) Saturated soil


(b) Dry soil

**Fig. 5.** Shear stress–angular displacement curves for the landslide soil at a shear rate

of 5°/min and different axial stresses for (a) saturated soil and (b) dry soil.

The residual strength envelope of the soil can be illustrated by plotting the shear

stress against axial stress, as shown in Fig. 6.



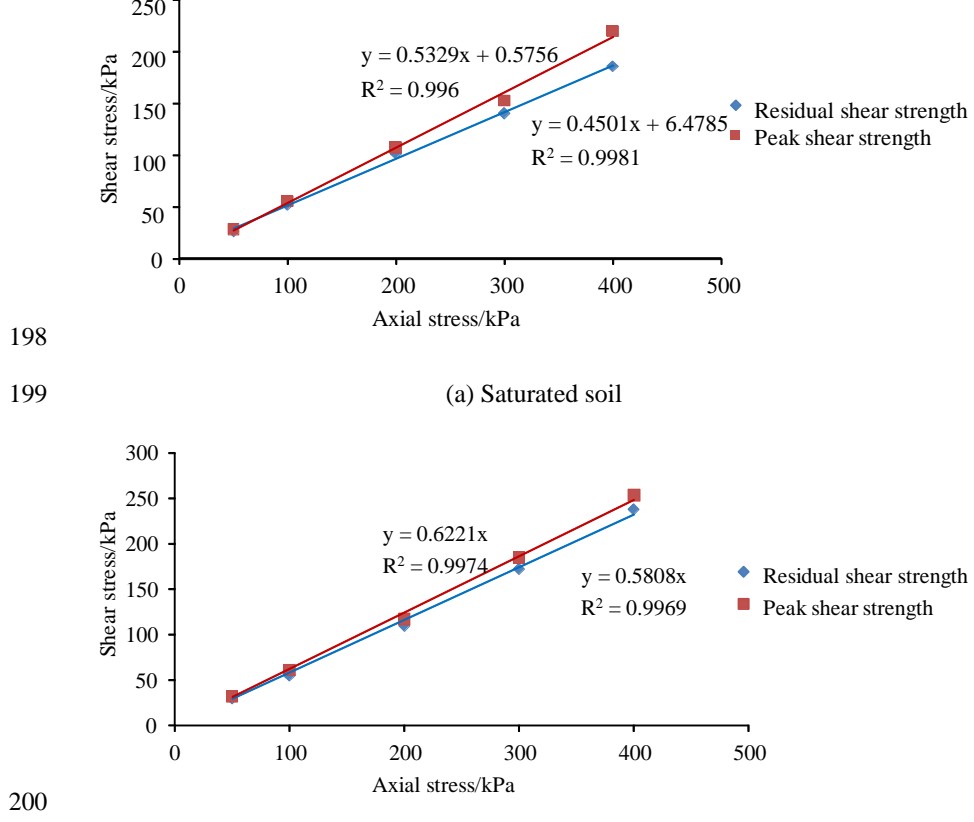


(a) Saturated soil

(b) Dry soil
**Fig. 6.** Residual strength envelopes for the landslide soils; (a) saturated soils, (b) dry
soils.
Based on Coulomb's equation, the peak and residual shear strengths of the
landslide soil were obtained and are listed in Table 3. Because the main trigger for the
Dafushan landslide was heavy rain, the residual strength of saturated soil is used for
the numerical simulation presented in Section 4 of this paper.
**Table 3** Cohesion and internal friction for landslide soils at peak and residual shear
strengths calculated from the Coulomb (Mohr-Coulomb) equation.



| Soil | Peak shear strength | | Residual shear strength | |
|---|---|---|---|---|
| | Cohesion $c_r$/kPa | Internal friction angle $\varphi_r$/° | Cohesion $c_r$/kPa | Internal friction angle $\varphi_r$/° |
| Saturated soil | 0.58 | 28.05 | 6.48 | 24.23 |
| Dry soil | 0 | 31.89 | 0 | 30.15 |

210 (2) Shear rate

211 Figure 7 shows the relationships between shear stress and angular deformation

212 under a normal stress of 200 kPa at shear rates of 1, 5, 10, and 20 °/min. As the shear

213 rate increases, the residual shear strengths increase slightly but the peak shear

214 strengths show the opposite reaction. However, the angular displacements at peak

215 shear strength increase significantly, as shown in Table 4.

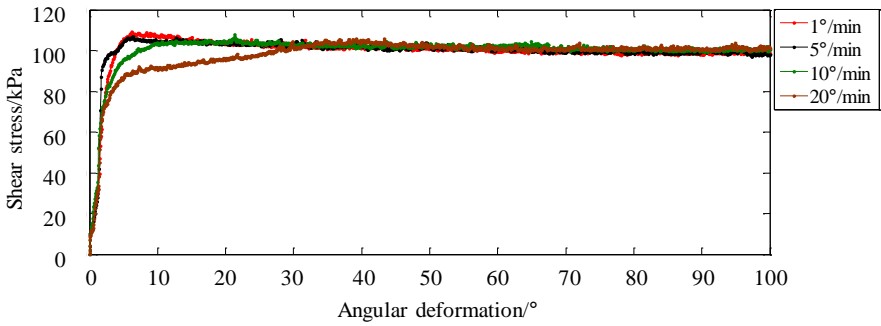

217 **Fig. 7.** Shear stress–angular displacement curves for saturated landslide soil under

218 200 kPa axial stress.

219 **Table 4** Differences in shear strengths and angular displacements for saturated

220 landslide soil at different shearing rates.



| Shearing rate (°/min) | Peak shear strength (kPa) | Residual shear strength (kPa) | Difference between peak and residual shear strength (kPa) | Angular displacement at peak shear strength (°) |
|---|---|---|---|---|
| 1 | 109.10 | 99.35 | 9.75 | 6.264 |
| 5 | 107.00 | 99.52 | 7.48 | 6.444 |
| 10 | 105.00 | 100.55 | 4.45 | 16.992 |
| 20 | 105.80 | 100.99 | 4.81 | 39.168 |

To analyze the relationship between the residual shear strength of the saturated
soil and the shear strain rate, the residual shear stress–shear strain rate curve can be
drawn (Fig. 8). The formula for calculating the shear strain rate is:
$$\dot{\gamma} = \frac{R\omega}{H}$$
(1)

where $\dot{\gamma}$ is the shear strain rate, $R$ is the average radius of the sample, $\omega$ is the
angular velocity, $H$ is the sample height.
As shown in Fig. 8, the residual shear strength of the saturated soil as determined
by these experiments increases linearly with shear strain rate. This result agrees with
the results reported by Li et al. (2013) and Dai et al. (2016). This relationship is
similar to the behavior of a viscous fluid and can be expressed by Eq. (2):
$$\tau = \eta\dot{\gamma} + f(\sigma)$$
(2)

where $\tau$ is shear stress, $\eta$ is the coefficient of viscosity. The intercept $f(\sigma)$
represents the shear stress when the shear strain rate equals 0.



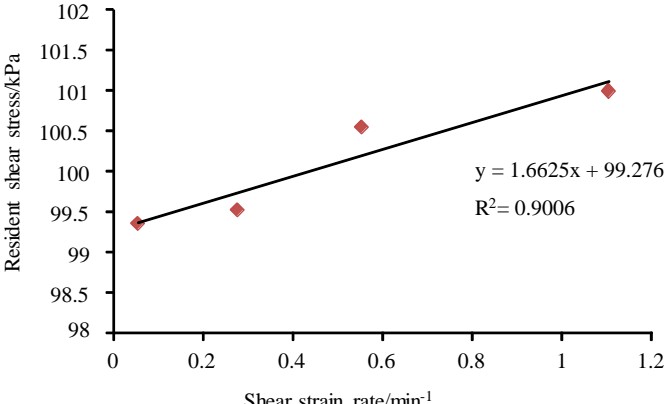


**Fig. 8.** Residual shear stress–shear strain rate curves for the saturated landslide soil.

**4. SPH-based numerical simulation for landslides**

*4.1 Calculation principles and SPH process methods*

*(1) Basic SPH concepts*

Smoothed particle hydrodynamics is a mesh-free and fully Lagrangian method based

on fluid dynamics. In Lagrangian models, the coordinates move with the medium

being modeled. The continuous medium is discretized into a series of arbitrarily

distributed discrete elements (called particles) and field variables (like energy,

velocity, density, or any other variable) for each particle can be calculated in the form

of SPH (Dao et al. 2013; Huang and Dai 2014).

The SPH method is built on interpolation theory with two essential

approximations. These approximations are smoothing and the particle (Huang et al.

2014). The smoothing approximation, also known as kernel approximation, describes

a function in a continuous form as an integral representation. The particle

approximation means that the value of a function for a particle can be determined by



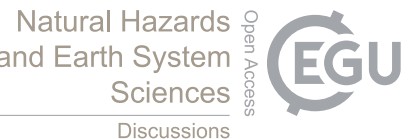

the average value of all the particles in the support domain. The smoothing and the
particle approximations can be expressed, respectively, by the following two
equations:

$$\langle f(x) \rangle = \int_{\Omega} f(x') W(x-x',h) dx' \tag{3}$$

$$\langle f(x) \rangle = \sum_{j=1}^{N} m_j \frac{f_j(x)}{\rho_j} W(x-x',h) \tag{4}$$

where the angle brackets represent a kernel approximation, $x$ is the location
vector of the particle, $x'$ denotes neighboring particle in the support area, $W$ is the
smoothing function, $h$ stands for the smoothing length, $\Omega$ stands for the volume of the
integral that contains $x$, $m$ is the mass, and $\rho$ is the density, $N$ is the total number of
particles.
*(2) Governing equations*
The Navier–Stokes equations in a computational fluid dynamics framework are used
as governing equations in this study. The equations of continuity and motion in the
SPH version can be expressed as:

$$\frac{d\rho_i}{dt} = \sum_{j=1}^{N} m_j \left( u_i^{\beta} - u_j^{\beta} \right) \frac{\partial W_{ij}}{\partial x_i^{\beta}} \tag{5}$$

$$\frac{du_i^{\alpha}}{dt} = \sum_{j=1}^{N} m_j \left[ \frac{\sigma_i^{\alpha\beta}}{(\rho_i)^2} + \frac{\sigma_j^{\alpha\beta}}{(\rho_j)^2} \right] \frac{\partial W_{ij}}{\partial x_j^{\beta}} + F_i \tag{6}$$

where $W_{ij}$ represents the smoothing function of particle $I$ calculated at particle $j$, $t$
is time, $u$ denotes the velocity vector, $\sigma$ is the stress tensor, $F$ represents the vector of
external force, and $\alpha$ and $\beta$ are the coordinate directions.
*(3) Model for a landslide simulation*





The Bingham model has been proved as one of the most effective models for runout
simulation of flowlike landslides (Marr et al. 2002; Moriguchi et al. 2009). In this
paper, the Bingham flow model is also adopted as the constitutive model for the
Dafushan landslide in this study. The relationship between shear stress and strain rate
can be written as:
$$\tau = \eta\dot{\gamma} + \tau_y.$$    (7)

Equation (8) can be modified by combining it with the Mohr-Coulomb yield

criterion to yield (Moriguchi et al. 2009):
$$\tau = \eta\dot{\gamma} + \sigma\tan\varphi + c$$    (8)

where $\tau$ denotes the shear stress, $\eta$ and $\tau_y$ represent the Bingham yield viscosity

and stress, respectively, $\dot{\gamma}$ is the shear strain rate, $\sigma$ is the pressure, $\varphi$ is the friction
angle, and $c$ is the cohesion.

For this study, the concept of equivalent viscosity was adopted to better integrate

the Bingham model into the SPH framework. The equivalent viscosity can be
expressed as:
$$\eta' = \eta + \tau_y/\dot{\gamma}.$$    (9)

The maximum value was defined by Uzuoka et al. (1998) as:

$$\eta' = \eta_0 + \frac{\tau_y}{\dot{\gamma}} \qquad \text{when} \quad \eta' < \eta_{\max}$$    (10)
$$\eta' = \eta_{\max} \qquad \text{when} \quad \eta' > \eta_{\max}$$    (11)

where $\eta_{\max}$ is the maximum value of $\eta'$.

*(4) Procedure for the numerical simulation*



A flow chart for the SPH numerical simulation is shown as Fig. 9. Details about how
the calculations are carried out can be found in Huang et al. 2014. The accuracy of
SPH program in landslide modelling was also fully validated in Huang et al. 2014.

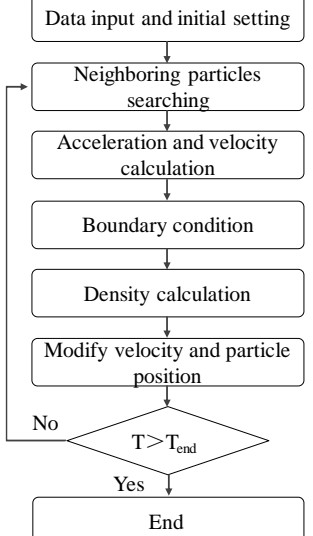


**Fig. 9.** Flow chart for the SPH numerical simulation used in this study.

*4.2 Dafushan landslide SPH simulation and results*

Based on a terrain model derived from an unmanned aerial vehicle and

structure-from-motion (Yu et al. 2017), an SPH simulation of the failure process of
the Dafushan landslide was conducted. This simulation was used to assess the
landslide's effects when failure occur. The numerical model is calculated on the basis
of a total of 3,242 particles, 1,537 particles for the slide mass and 1,705 for the fixed
boundary. Figure 10 is a longitudinal section of the model slide with the particles in
the slide mass shown in red, the boundary particles shown in blue. The diameter of

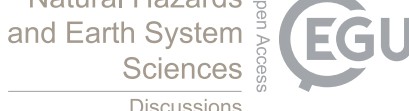



each particle is 0.5 m. The soil particles in the model can be deformed in both the
vertical and horizontal directions under gravitational force in the vertical direction.

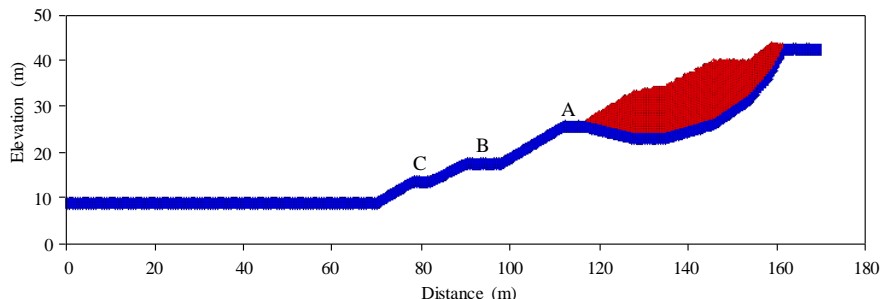


**Fig. 10.** Longitudinal section of the SPH numerical model of the Dafushan landslide.
The particles representing the slide mass are shown in red, the particles representing

the fixed boundary are shown in blue.

Table 5 lists the parameters used in the SPH simulation of the landslide. The

shear strength parameters listed in Table 5, $c$ and $\varphi$, are the values calculated from the
ring shear tests (Table 3).

**Table 5** Parameters used in the SPH simulation of the Dafushan landslide.

| Density $\rho$ (kg/m$^3$) | 1770 |
|---|---|
| Residual cohesion $c$ (kPa) | 6.48 |
| Residual internal friction Angle $\varphi$ (°) | 24.23 |
| Acceleration of gravity $g$ (m/s$^2$) | 9.80 |
| Unit time step $\Delta t$ (s) | 0.003 |
| Time step ($n$) | 40000 |

Figures 11(a)-11(g) show the flow process of Dafushan landslide predicted by

the SPH simulation. In Fig. 11, the solid black line represents the bed on which the





mass slides, the red dashed line represents the SPH-modeled ground surface. At time t
= 0, this red line is the ground surface before slide failure. For times after t = 0, it is
the top surface of the flowing mass of soil that constitutes the moving landslide mass
as predicted by the SPH simulation results. In the model, the time the failed Dafushan
landslide lasts, from initiation to the whole landslide mass coming to rest, is 120 s.
The model predicts that the landslide would cut the fire road at t = 12 s and cover the
expressway at t = 36 s. When the landslide stops sliding at 120 s, slide material would
cover about a 25 m wide swath of ground beyond the foot of the topographic slope.

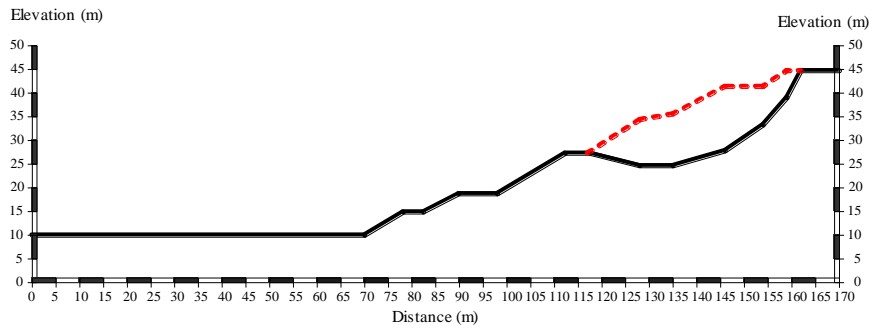


(a) t = 0 s

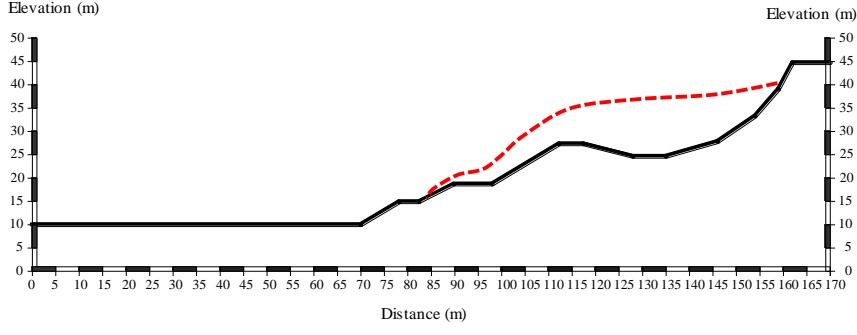


(b) t = 12 s




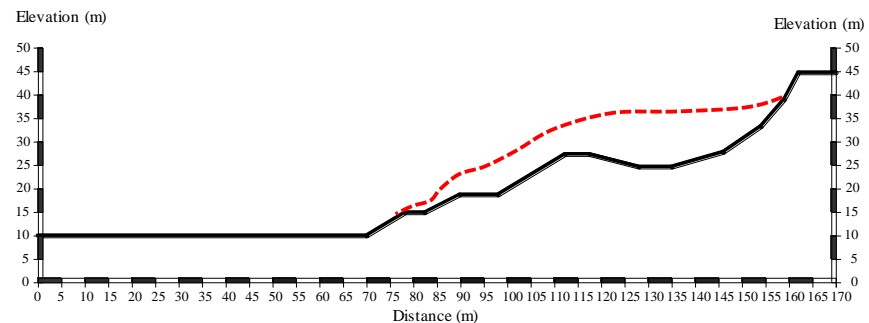


(c) t = 24 s

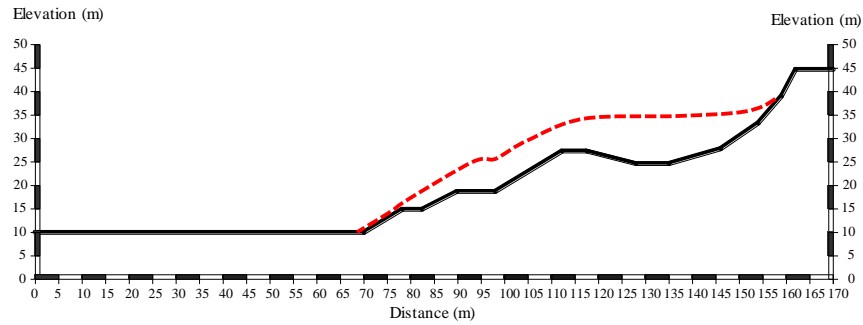


(d) t = 36 s

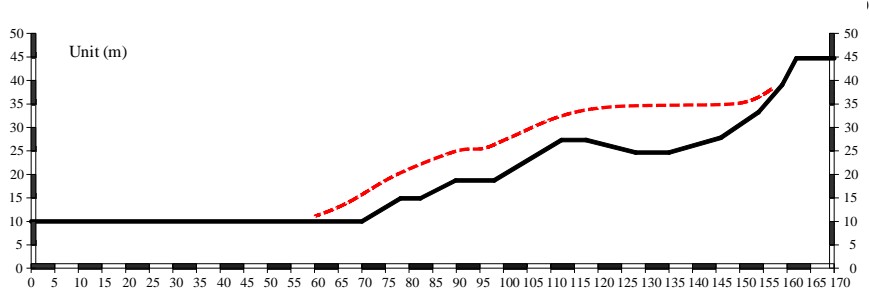





(e) t = 60 s


(f) t = 84 s


(g) t = 120 s

**Fig. 11.** Longitudinal profiles showing the results of the SPH forecasting model. The
panels represent the outline of the Dafushan landslide from the time the slide is



initiated at t = 0 s (panel a) through the slide finally coming to rest at t = 120 s (panel

342                                          g).

Because this SPH simulation is a Lagrangian method, it can track the velocity

and displacement of each particle accurately. The velocity and displacement curves
for the front and rear edges of simulated landslide are shown in Figs. 12 and 13. As
shown in Fig. 12, the velocity of the front edge increases rapidly after slope failure
begins and reaches three velocity peaks as the slide passes the three steps labeled A, B,
and C shown in Fig. 10. The speed of the front and the times after initiation that it
reaches these three steps are 5.23 m/s at 0.6 s at step A, 6.66 m/s at 9.3 s at step B,
and 1.92 m/s at 23.6 s at step C. Unlike the front edge of the landslide, the velocity of
the landslide's rear edge shows only a single peak. The maximum speed is 1.40 m/s;
this appears 3.8 s after the slide is initiated.

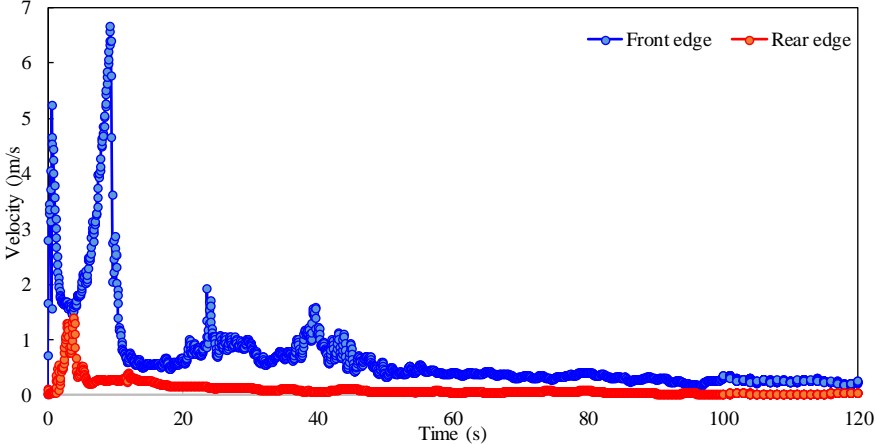


**Fig. 12.** Velocity curve of the front and rear edge of Dafushan landslide as predicted

by the SPH model.



According to the Fig. 13, the maximum flow distances of the front and rear edge
are up to 82 m and 12.3 m, respectively. The front edge of the slide will destroy the
fire road about 10–12 s after the slide starts and reach the highway at t = 36 s.
Thereafter, the velocity gradually approaches zero as the flow distance increases. The
maximum distance the landslide flows is approximately 82 m, and the speed of the
flow can be divided into three stages. The flow is fastest from 0–10 s, slower from
10–45 s, and relatively slow from 45–120 s. However, once signs of failure are
observed at the Dafushan landslide site, evacuation of personnel and vehicles within
about 25 m of the slope should begin immediately.

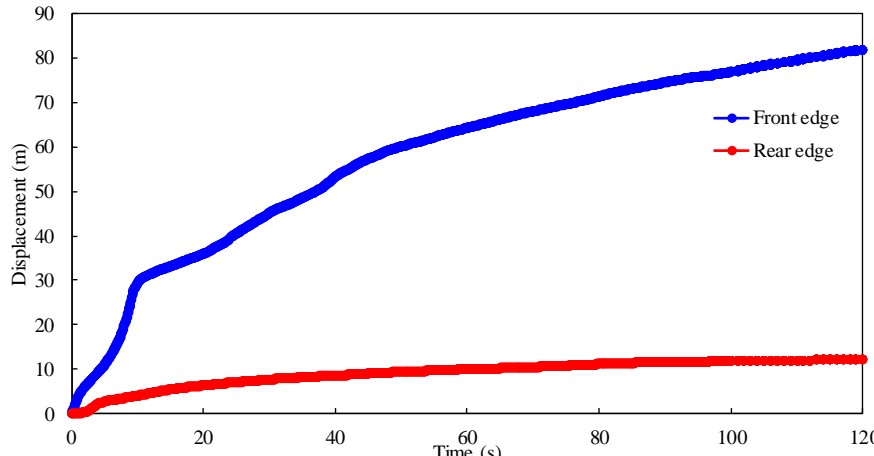


**Fig. 13.** Displacement curve of the front and rear edge of Dafushan landslide as

predicted by the SPH model.

**5. Conclusions**
In this study, the SPH method incorporating soil mechanical parameters derived from
ring shear tests is used to predict the flow of a potential landslide that could develop



on an unstable slope in Guangzhou City, China. This study provides basic information
for landside disaster mitigation. The conclusions are:

(1) Under the same shear rate, soil shear strength increases with increasing axial

stress. For the conditions used in this study, under high axial stress (> 200 kPa) the
soil exhibits strain softening.

(2) During ring shear tests, as the shear rate increases, the residual shear

strengths increase slightly but the peak shear strengths decrease as the angular
displacements at peak shear strength increase significantly.

(3) A SPH-based numerical simulation of the potential Dafushan landslide

conducted to predict the scope of the landslide and track the slide velocity at different
times shows that the landslide would cut the fire road at t = 12 s and cover the
expressway at t = 36 s. And once signs of failure are observed at the Dafushan
landslide site, evacuation of personnel and vehicles within about 25 m of the slope
should begin immediately.
**Acknowledgements**
This work was supported by the National Science Fund for Distinguished Young
Scholars of China (Grant No. 41625011), the Fundamental Research Funds for
National University, China University of Geosciences (Wuhan) (Grant No.
CUGL170806), and the National Key Technologies R&D Program of China (Grant
No. 2012BAJ11B04).

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
