# Peer review of "Forecasting landslide mobility using a SPH model and ring shear strength tests: A case study"

_Natural Hazards and Earth System Sciences, 2018_

## Referee Comment (RC1) · Anonymous Referee #1 · 9 Feb 2018

This manuscript presents a SPH simulation on a flowlike landslide. A series of ring shear tests under different axial stresses and shear velocities were conducted to obtain the soil strength parameters which were necessary in the numerical simulation. My detail comments are listed as follows: 1.In Figure 7, no obvious relationship between shear stress and shear strain rate is found. 2.What's the new contribution on the SPH model should be highlighted in this work. 3.In this numerical simulation, 3242 particles would establish a poor spatial discretization. An analysis of the discretization error related to the particle distance is strongly encouraged. 4.How to decide the time step in the simulation? A convergence analysis is suggested. 5.Ring shear tests were conducted to evaluate the residual shear strength of slip zones, but in the simulation, the

authors used the strength parameters to describe the behavior of the whole landslide body. 6.According to the numerical results, the maximum velocity of the landslide is 6.66 m/s. However, in the ring shear tests, the maximum shear rate is 20 °/min, which is much smaller than the numerical result. So can the strength parameters obtained from the tests be applied in the numerical simulation? 7.According to the Figure 13, it seems that the landslide is still moving at 120s after failure, see the blue line.

---

## Referee Comment (RC2) · Anonymous Referee #2 · 21 Mar 2018

The study aims to forecasting the mobility of a landslide in Guangzhou city using a SPH model and ring shear strength tests. In order to address the question, the author carried on a series of work, such as filed investigation, ring shear strength tests, and numerical simulation. Overall, the manuscript is in a good manner presentation and show a detail and valuable case study on landslide mobility forecasting. The special comment include: (1) Title: Delete "post-failure", "data from" (2) Please provide a plan map include topographic information of the landslides. Put all the important points, such as sampling location, the profile position shown in fig.1d. (3) Please explain how the Longitudinal geologic section be drawn without drilling hole.

---

## Author Comment (AC1) · 30 Oct 2018

**RC: In Figure 7, no obvious relationship between shear stress and shear strain rate is found.**

AC: Thank you very much for the referees' comment. Because the difference between the test results is very slight, so the relationship in the Figure 7 is not very obvious. Hence, we calculated from the experimental data and the specific data is shown as Table 4. As the shear rate increases, the residual shear strengths increase slightly but the peak shear strengths show the opposite reaction. However, the angular displacements at peak shear strength increase significantly.

**Table 4** Differences in shear strengths and angular displacements for saturated landslide soil at different shearing rates.

| Shearing rate (°/min) | Peak shear strength (kPa) | Residual shear strength (kPa) | Difference between peak and residual shear strength (kPa) | Angular displacement at peak shear strength (°) |
|---|---|---|---|---|
| 1 | 109.10 | 99.35 | 9.75 | 6.264 |
| 5 | 107.00 | 99.52 | 7.48 | 6.444 |
| 10 | 105.00 | 100.55 | 4.45 | 16.992 |
| 20 | 105.80 | 100.99 | 4.81 | 39.168 |

**RC: What's the new contribution on the SPH model should be highlighted in this work.**

AC: We sincerely appreciate the referees' comment. The case study is a previously identified unstable slope. Once the slope failure, it would induce serious damages. So, it is urgent to predict the slide distance once it slides as the rainy season is coming. Previous studies have been conducted on the verification analysis of landslide, which prove the accuracy of SPH model is relatively high (Huang et al. 2012; Hu et al. 2015). The error of slide distance can reach 4.6% (Huang et al. 2012). While, this study extends the application of SPH model to predictive analysis of unstable landslides. And accurate soil parameters derived from ring shear tests are introduced in the calculation. In addition, considering the referees' comment, two sets of comparative calculation were carried out, which demonstrate the robustness of the SPH method. We have highlighted the contribution in the manuscript.

-*Huang, Y. et al. (2012). Run-out analysis of flow-like landslides triggered by the Ms 8.0 2008 Wenchuan earthquake using smoothed particle hydrodynamics. Landslides, 9(2), 275-283.*

-*Hu, M. et al. (2015). Three-dimensional run-out analysis and prediction of flow-like landslides using smoothed particle hydrodynamics. Environ Earth Sci, 73(4), 1629-1640.*

**RC: In this numerical simulation, 3242 particles would establish a poor spatial discretization. An analysis of the discretization error related to the particle distance is strongly encouraged.**

AC: We sincerely appreciate the referees' comment. The study about discretization error related to the particle distance is very valuable. But in our case, the landslide is unstable but not failure yet, so there are no real data (like slide distance) to evaluate the error related to particle distance. Hence, the particle distance in our study is based on our computing experience and previous studies. For example, in reference Cuomo et al. (2016), only 639 points in the source area were used; In reference Huang et al. (2013), three are two cases included 2619 particles in total (1653 particles for the slide) and 2188 particles in total (1260 particles for the slide), respectively. And in reference Huang et al. (2013) and Cuomo et al. (2016), the SPH results show high degree of similarity with surveyed ones. In addition, their calculation range is much larger than ours, hence 3242 particles for 170 m in our study should be enough.

In addition, we actually did two different calculations with different parameters, as shown in the following table and figure. The parameters of computer are 3.40 GHz CPU and 16 GB RAM. The results have a high degree of similarity in the slide distance, but the time consuming is about 2.76 times than before. Hence, 3242 particles with diameter of 0.5 m and time step of 0.003 are applied in our study. This part has been added in the manuscript, including the table and figure.

*-Cuomo S, Pastor M, Capobianco V, Cascini L (2016) Modelling the space–time evolution of bed entrainment for flow-like landslides. Engineering Geology 212: 10–20*

*-Huang Y, Dai Z, Zhang WJ, Huang MS (2013) SPH-based numerical simulations of flow slides in municipal solid waste landfills. Waste Management & Research, 31(3): 256–264*

Table 6 Predicted results in Dafushan landslide with different calculating parameters

| Case | Particle diameter (m) | Particle number | Time step (s) | X-coordinate of the slide front (m) | Time consuming (min) |
|---|---|---|---|---|---|
| 1 | 0.5 | 3242 | 0.003 | 32.43 | 66 |
| 2 | 0.4 | 4471 | 0.002 | 33.45 | 182 |

[Figure]

Fig. 11. Predicted results in Dafushan landslide with different calculating parameters

***RC: How to decide the time step in the simulation? A convergence analysis is suggested.***

AC: We sincerely appreciate the referees' comment. We used 0.003 as the time step for three reasons as follow.

--According to the reference Cummins and Rudman (1999), the time step can be determined by the CFL stability constraint. In this study, $|u|_{max}$ is estimated as 10 m/s, so $\Delta t \le 0.0125$.

$$\Delta t \le 0.25 \frac{h}{|u|_{max}}$$

Where $h$ is the particle diameter, $|u|_{max}$ is anticipated maximum particle velocity in the computation.

--The previous published papers about the application of SPH in flow slides are summarized as follows. When the time step is 0.0025~0.03, the calculated results agree with the survey results well.

| Reference | Time step (s) |
|---|---|
| SPH-based simulation of flow process of a landslide at Hongao landfill in China. Natural Hazards, 2018, 93(3): 1113-1126. | 0.005 |
| SPH-based numerical simulation of catastrophic debris flows after the 2008 Wenchuan earthquake. Bulletin of Engineering Geology and the Environment, 2015, 74(4): 1137-1151. | 0.0025; 0.03 |
| SPH-based numerical simulations of flow slides in municipal solid waste landfills. Waste Management & Research, 2013, 31(3): 256-264. | 0.03 |

-- Based on previous studies and repeated trials, we conducted two different

calculations with different sets of parameters, and found that it is convergent when the time step is 0.003 s.

| Case | Particle diameter (m) | Particle number | Time step (s) | X-coordinate of the slide front (m) | Time consuming (min) |
|---|---|---|---|---|---|
| 1 | 0.5 | 3242 | 0.003 | 32.43 | 66 |
| 2 | 0.4 | 4471 | 0.002 | 33.45 | 182 |

**RC: Ring shear tests were conducted to evaluate the residual shear strength of slip zones, but in the simulation, the authors used the strength parameters to describe the behavior of the whole landslide body.**

AC: Thank you very much for the referees' comment. Actually, the slip zone and the landslide body are the same soil layer, viz., the first soil layer. For ease of understanding, the sliding surface were added in the Fig. 2 as follows, which is also modified considering another referees' comment. In addition, in the ring test the reconstituted soil sample was tested with the considering of different normal stresses. Hence, the parameters calculated from the ring shear tests can be used for the whole slide part.

[Figure]

**Fig. 2.** Geology and soil at the Dafushan landslide. (a) Longitudinal geologic section of the unstable slope shown in Fig. 1(c). (b) Photograph of the silty clay landslide soil.

**RC: According to the numerical results, the maximum velocity of the landslide is 6.66 m/s. However, in the ring shear tests, the maximum shear rate is 20 /min, which is much smaller than the numerical result. So can the strength parameters obtained from the tests be applied in the numerical simulation?**

AC: Thank you very much for the referees' comment. Your question is very valuable, which is also a difficult issue in the current research. At present almost all the ring shear test apparatus cannot reach the shear rate of actual landslide in the failure process, except very few self-developed test systems. For example, K. Sassa et al. at the Disaster Prevention Research Institute, Kyoto University developed a ring shear apparatus can reach a maximum speed of 3m/s. In this study, the purpose of the ring shear test is to

obtain the law of soil strength and some reliable calculation parameters. In addition, when the shear rate is high, the soil sample will be squeezed out from the shear box, which will affect the accuracy of the results. Hence, in our study, the tests were carried out within the reliable shear velocity range of the ring shear test system.

**RC: According to the Figure 13, it seems that the landslide is still moving at 120s after failure, see the blue line.**

AC: We sincerely appreciate the referees' comment. We increased the computing time and recalculated, and the conclusion was drawn that the landslide stopped moving at 200 s.

[Figure]

(h) t = 200s

**Fig. 12.** Longitudinal profiles showing the results of the SPH forecasting model. The panels represent the outline of the Dafushan landslide from the time the slide is initiated at t = 0 s (panel a) through the slide finally coming to rest at t = 200 s (panel h).

[Figure]

**Fig. 13.** Velocity curve of the front and rear edge of Dafushan landslide as predicted by the SPH model.

[Figure]

**Fig. 14.** Displacement curve of the front and rear edge of Dafushan landslide as predicted by the SPH model.

---

## Author Comment (AC2) · 30 Oct 2018

**RC: Title: Delete "post-failure", "data from".**

AC: Thank you very much for the referees' comment.

The title has been modified as "*Forecasting landslide mobility using a SPH model and ring shear strength tests: A case study*".

*RC: Please provide a plan map include topographic information of the landslides. Put all the important points, such as sampling location, the profile position shown in fig.1d.*

AC: We sincerely appreciate the referees' comment. The Figure 1 has been modified as follows.

[Figure]

**Fig. 1.** Overview of the Dafushan landslide. (a) Landslide location; (b) Geomorphologic and geologic map of the landslide area; (c) Aerial view of the unstable slope; (d) Engineering activities on the slope (modified based on Yu et al. (2017) with permission of Springer).

*RC: Please explain how the Longitudinal geologic section be drawn without drilling hole.*

AC: Thank you very much for the referees' comment. The drilling holes has been

marked on the Fig.1 and Fig. 2, as follows. After the modified, the figures are more cleared and consistent with the text.

[Figure]

**Fig. 2.** Geology and soil at the Dafushan landslide. (a) Longitudinal geologic section of the unstable slope shown in Fig. 1(c). (b) Photograph of the silty clay landslide soil.